# An Operational DNA Strand Displacement Encryption Approach

**DOI:** 10.3390/nano12050877

**Published:** 2022-03-06

**Authors:** Enqiang Zhu, Xianhang Luo, Chanjuan Liu, Congzhou Chen

**Affiliations:** 1Institute of Computing Science and Technology, Guangzhou University, Guangzhou 510006, China; zhuenqiang@gzhu.edu.cn (E.Z.); 2112006164@e.gzhu.edu.cn (X.L.); 2School of Computer Science and Technology, Dalian University of Technology, Dalian 116024, China; 3School of Electronics Engineering and Computer Science, Peking University, Beijing 100871, China; chencongzhou@pku.edu.cn

**Keywords:** DNA strand displacement reaction, DNA encryption, huffman coding

## Abstract

DeoxyriboNucleic Acid (DNA) encryption is a new encryption method that appeared along with the research of DNA nanotechnology in recent years. Due to the complexity of biology in DNA nanotechnology, DNA encryption brings in an additional difficulty in deciphering and, thus, can enhance information security. As a new approach in DNA nanotechnology, DNA strand displacement has particular advantages such as being enzyme free and self-assembly. However, the existing research on DNA-strand-displacement-based encryption has mostly stayed at a theoretical or simulation stage. To this end, this paper proposes a new DNA-strand-displacement-based encryption framework. This encryption framework involves three main strategies. The first strategy was a tri-phase conversion from plaintext to DNA sequences according to a Huffman-coding-based transformation rule, which enhances the concealment of the information. The second strategy was the development of DNA strand displacement molecular modules, which produce the initial key for information encryption. The third strategy was a cyclic-shift-based operation to extend the initial key long enough, and thus increase the deciphering difficulty. The results of simulation and biological experiments demonstrated the feasibility of our scheme for encryption. The approach was further validated in terms of the key sensitivity, key space, and statistic characteristic. Our encryption framework provides a potential way to realize DNA-strand-displacement-based encryption via biological experiments and promotes the research on DNA-strand-displacement-based encryption.

## 1. Introduction

Over the past few years, the world has seen a stunning transformation in how information is exchanged. Communication online (through various platforms) has gradually become an indispensable means for information exchange, and ensuring data security has become one of the most concerning problems.

Cryptography plays a pivotal role in protecting the security of data communication by transforming plaintexts into unrecognizable codes [1,2]. Conventional cryptography, which depends excessively on the high computational complexity of mathematical calculations, is facing increasing risks of encryption cracking as computing capabilities are rising. Therefore, new encryption methods have been increasingly studied. As a nanomaterial, DeoxyriboNucleic Acid (DNA) can store a large amount of information, and with the rapid development of nanotechnology, DNA nanotechnology has been widely studied for encryption. DNA encryption, as a novel technique of cryptography, was proposed by Gehani et al. [3]. In DNA encryption, data are protected by transforming them into digital DNA codes. Because of the exclusive advantages of DNA molecules, including their large scale of parallelism, high storage capacity, and low power consumption, it is widely believed that DNA encryption can work with huge data and can potentially increases information security [4].

There is a growing body of literature recognizing the importance of DNA encryption. To solve the storage problem of one-time pad, Gehani et al. [3] first designed a one-time-pad-based DNA encryption program. In 2012, Wang et al. [5] proposed a new one-time one-key encryption algorithm based on the ergodicity of the skew tent chaotic graph. In 2014, Mokhtar et al. [6] combined a chaotic system with DNA coding to design a one-time pad encryption scheme. In [7], Yang et al. proposed a one-time pad encryption device based on DNA self-assembly technology. Because the keys generated in one-time pad approaches are not reusable, it is difficult to produce enough keys for encryption. A common method to address this problem is code transformation (i.e., transforming (0,1)-sequences into DNA sequences). In 2012, Liu et al. [8] proposed an image encryption method by means of a novel confusion and diffusion method, in which a DNA complementary rule was designed to confuse the pixels. To enhance the degree of confusing the pixels, Rehman et al. [9] in 2014 proposed a new gray image block cipher, which dynamically selects a rule from newly designed DNA complementary rules to encode and decode each pixel in a block. In 2016, based on the combination of the dynamic S-box and chaotic systems, Liu et al. [10] proposed a new image encryption scheme and showed that the proposed algorithm can reduce the correlation coefficients of images in three directions. In 2018, Wu et al. [11] designed a new chaotic mapping, called 2D-HSM. Then, they proposed an image encryption scheme combining 2D-HSM with DNA approaches and demonstrated its excellent performance. In [12,13], the authors employed chaotic series generated by a chaotic system to randomly select the coding rules, by which the security of encryption can be improved significantly. More recently, Wang et al. [14] proposed an image encryption algorithm based on ladder scrambling and DNA coding, which has a lower correlation of images compared to previous algorithms. In addition, some studies have attempted to improve the security of DNA encryption by performing operations on DNA codes, such as Addition (ADD) [15,16], Subtraction (SUB) [15,16], Exclusive Or (XOR) [16,17,18], and Exclusive Nor (XNOR) [18].

DNA encryption has been extensively studied along with the research on DNA nanotechnology in recent years. Due to the biological complexity of DNA nanotechnology, DNA encryption brings in the additional difficulty of deciphering, and thus can enhance information security. As a new approach in dynamic DNA nanotechnology, DNA Strand Displacement Reaction (SDR) has particular advantages such as being enzyme free and self-assembly. SDR has attracted considerable attention in recent years and has been widely applied to build various molecular systems [19] (it should be noted that the materials (DNA single strands) required for DNA strand displacement experiments are first designed by researchers, then commissioned to manufacture, and finally assembled into DNA molecules (complex structure)). A DNA SDR can be described as a molecular dynamic process (Figure 1), where a single-stranded DNA molecule is combined with a double-stranded DNA molecule through short complementary single-stranded DNA domains (called toeholds; see td and $td^{*}$), and a new stable double-stranded DNA molecule will be formed and a new single-stranded DNA molecule released from the original double strand. Notice that this can only happen gradually. Previous research has demonstrated that by designing appropriate DNA SDR, one can approximately realize all chemical reactions with ideal forms [20,21]. For example, in [22], SDR-based DNA switching circuits were designed for digital computing; in [23], the authors developed a time-sensitive molecular circuit based on SDR, called the cross-inhibitor, which can execute mutual inhibition; in [24,25], DNA strand displacement for microRNA detection was investigated; in [26], the authors analyzed the morphological manipulation of DNA gel microbeads with biomolecular stimuli by using SDR; in [27], the authors proposed an SDR-based chemical reaction network to solve 0–1 integer programming problems.

Designing encryption algorithms with the aid of DNA SDR has also been attempted. In [28], by using DNA SDR to extract secret keys, Zhang et al. proposed an image encryption algorithm on the basis of a chaos system. To obtain the keys with this approach, the DNA of the chains obtained by SDR must be sequenced. This may lead to decryption failures when current sequencing techniques are used. In [29], the authors designed six DNA SDR modules and combined them with the XOR operation to create a new encryption algorithm. Although the proposed algorithm may have a high capacity to resist statistical attacks, it relies heavily on real-time concentration detection. Therefore, it is still in a simulation stage and is difficult to realize via biological experiments because of the complicated design program.

During a DNA strand displacement experiment, it is difficult to monitor and detect the concentration of the target DNA strand in real time, and the changes in the design of the DNA sequence can easily lead to changes in the reaction rate. For these reasons, the study of DNA-strand-displacement-based encryption is still in the theoretical or simulation stage. To facilitate the implementation of DNA encryption via the biological experiment of DNA strand displacement, we introduced in this work a novel bio-experiment-based encryption framework. In this approach, three strategies were adopted, including a Huffman-coding-based transformation rule to confuse the plaintext, two SDR-based molecular modules to generate the initial key, and a cyclic-shift-based mechanism to extend and confuse the key. Note that most studies on DNA encryption techniques focus mainly on how to design complex rules to hide confidential information in DNA codes, without considering whether the designed scheme can be realized by biochemical experiments. Our approach enhances the feasibility of biochemical experiments and reveals two advantages. First, it improves the security of key transmission. To obtain the keys, one has to perform biochemical experiments, for which the results are sensitive to various conditions, such as temperature, time, and concentration. Therefore, our approach provides excellent protection against decoding. Second, it combines biochemical experiments with other techniques such as code transformation, which generates a new confusion and diffusion method to create a secure cipher, thus enhancing the cipher strength.

In order to verify the feasibility of the proposed approach, we first present an encryption example. Then, we refer to [29] for the analysis of its performance in encryption in terms of three aspects, viz., key sensitivity, key space, and statistic characteristics. Note that a good encryption method should be sensitive to the key, that is, when the key changes slightly, the encryption and decryption results will be sufficiently different. Meanwhile, a good encryption method should also have a large key space to resist brute force attacks. Besides, we also analyzed the statistical characteristic of our approach to demonstrate that it can cope with statistical attacks. Our encryption framework provides a potential way to realize DNA-strand-displacement-based encryption via biological experiments and promotes the research on DNA-strand-displacement-based encryption.

The remainder of the paper is organized as follows. Section 2 introduces the encryption framework and the process of the encryption algorithm. Section 3 presents the experimental validation of the feasibility of our approach by designing specific modular reactions. In Section 4, we analyze the performance of our approach in encryption security. The results imply that the proposed scheme is sensitive to the keys and possesses high resistance against statistical attacks. Finally, a summary of the main findings, along with some discussion and concluding remarks are provided in Section 5.

## 2. A Bio-Experiment-Based Encryption Approach

### 2.1. Encryption Framework

In view of the increasing need for dealing with large data and ensuring data security, we propose a novel bio-experiment-based DNA encryption method based on the DNA strand displacement technique. In this section, we first present the framework of our encryption method (Algorithm 1).
**Algorithm 1:** A new bio-experiment-based encryption framework.
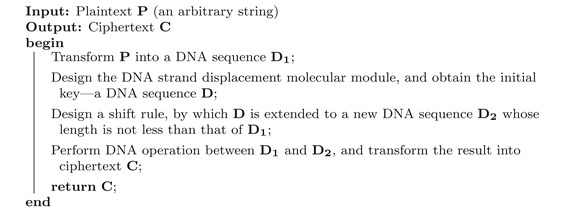


The encryption starts with a plaintext input P, i.e., an arbitrary string, and transforms it into a DNA sequence $D_{1}$ (Line 2), which will be taken as a substrate in the subsequent DNA computation. To generate the DNA sequence key D by biochemical experiments, some digital seeds are first obtained by recording the state changes (such as fluorescence color change or concentration change) during a designed experiment (Line 3). The next step is to extend D (Line 4) to a new DNA sequence $D_{2}$ with a length at least that of $D_{1}$ for later use in DNA computation. Finally, it produces the desired ciphertext (Line 5) by performing DNA computations (such as XOR and ADD) between $D_{1}$ and $D_{2}$, together with some transformation strategies.

### 2.2. Huffman Coding and Data Transformation

Huffman coding is an efficient method for compressing data without losing information. By using this technique, Ailenberg and Rotstein [30] proposed a simple, but efficient coding method for information storage in DNA and showed its potential ability in coding DNA. Inspired by this, we designed a Huffman-coding-based method, called *tri-phase transformation* (TPT), to confuse P.

TPT first transforms P into a DNA sequence $P_{1}$ according to the rule listed in Appendix A; then, it transforms $P_{1}$ into a (0,1)-sequence $P_{2}$ via Huffman coding; finally, by using the rules listed in the first column in Appendix A, it transforms $P_{2}$ into a new DNA sequence $D_{1}$, which is an ingredient for subsequent DNA operations. Specifically, the process from $P_{1}$ to $P_{2}$ can be described as follows.

For each base x∈{A,T,G,C}, denote by ω(x) the weight of *x*, which is defined as the number of *x* that appear in $P_{1}$. Then, construct a Huffman binary tree with four leaves in the following way: select two bases with the smallest weights as two leaves, denoted by $x_{1}$ and $x_{2}$, where $\omega \left(x_{1}\right)\leq \omega \left(x_{2}\right)$, and add a new vertex $y_{1}$ joining $x_{1}$ and $x_{2}$ such that $x_{1}$ and $x_{2}$ are the left and the right children of $y_{1}$, respectively; set $\omega \left(y_{1}\right)=\omega \left(x_{1}\right)+\omega \left(x_{2}\right)$ select two elements from $\{A,G,C,T,y_{1}$}$\backslash \{x_{1},x_{2}\}$ with the smallest weights, denoted by $x_{3}$ and $x_{4}$, where $\omega \left(x_{3}\right)\leq \omega \left(x_{4}\right)$, and add a new vertex $y_{2}$ joining $x_{3}$ and $x_{4}$ such that $x_{3}$ and $x_{4}$ are the left and right children of $y_{2}$, respectively; set $\omega (y_{2})$ = $\omega \left(x_{3}\right)+\omega \left(x_{4}\right)$, and add a new vertex $y_{3}$ jointing $y_{2}$ and the element in $\{A,G,C,T,y_{1}$}$\backslash \{x_{1},x_{2},x_{3},x_{4}\}$ such that the one with the smaller weight is the left child of $y_{3}$ and the other is the right child of $y_{3}$. Now, for each edge xy of the constructed tree such that *y* is a child of *x*, assign weight zero to it if *y* is the left child of *x*, and assign weight one to it if *y* is the right child of *x*. As a result, each base (a leaf) can be encoded into a (0,1)-sequence, which subsequently appears in the edges of the path from the root to the leaf, and $P_{1}$ is encoded into a (0,1)-sequence *Z*. Observe that the length of *Z* may be an odd number. To transform *Z* into the DNA sequence $D_{1}$ according to Appendix A, we have to modify it to have an even length. Our approach was as follows: if *Z* has an even length, add 00 to *Z* at the end of *Z*; otherwise, add 101 to *Z*. As an example, we considered a DNA sequence TTCCAGCGGAC, for which ω(A)=2,ω(G)=3,ω(C)=4, and ω(T)=2. By constructing a Huffman tree, A is encoded into 000, G is encoded into 01, C is encoded into 1, and T is encoded into 001. As a result, TTCCAGCGGAC is encoded into *Z* = 0010011100001101010001. Since *Z* has an even length, 00 is added at the end of *Z* and $D_{1}$ = ACGTAATGGAGA.

As described in Algorithm 1, our approach depends on DNA operations to generate the final ciphertext. Two such operations, XOR and ADD, are used in our subsequently designed algorithm, where the rules of these two operations are shown in Appendix A, respectively.

### 2.3. SDR Modules and Seed Encoding

Let us now turn to the design of initial keys, which first generate seeds in the form of “2-1” or “1-2” for the keys via the corresponding SDR modules. Two SDR modules are used to encode these two seeds based on the concentration change of the main species before and after the strand displacement reactions (the concentration change of the species should be normalized to the form *p*-*q* such that both *p* and *q* are integers, i.e., $1-\frac{1}{2}$ should be replaced by 2:1).

#### 2.3.1. Degradation Reaction Module

The principle of this module is presented in Figure 2, and its mechanism can be described by the reactions listed in Equation (Equation 1).
(1)$$\left\{\begin{array}{c} A+B\overset{k_{1}}{⟶}W_{1}+W_{2}\\ A+D\overset{k_{2}}{⟶}E+F+W_{2}\\ G+E+F\overset{k_{3}}{⟶}A+W_{3}+W_{4} \end{array}\right.$$

The process of the reaction can be described as follows: this module mainly involves four initial species, including Single-stranded A and Complexes B, D, and G. We add the inputs A, B, D, and G into the biochemical reaction module simultaneously, and then a series of reactions is activated, after which the concentration of A is reduced to half of its original concentration, as shown in Equation (Equation 2). This is because A is consumed by both B and D and is generated by only one reaction (the third reaction listed in Equation (Equation 3). Specifically, the toehold $a_{5}$ of A binds to the domain $a_{5}^{*}$ of B (and also D), and then, branch migration moves gradually to domain $a_{1}$, which releases single-stranded $W_{1}$ (and Single-stranded E and F) together with double-stranded $W_{2}$. Furthermore, the toehold $s_{2}$ of E (and $t_{2}$ of F) binds to the domain $s_{2}^{*}$ (and $t_{2}^{*}$) of G, and then, branch migration moves gradually to the domain $a_{3}$ (and $a_{1}$), which releases the desired Single-stranded A and forms double strands $W_{3}$ and $W_{4}$. Observe that both A and G carry a dye at their $3^{\prime }$ end, and B, D, and G each carry a quencher at their $5^{\prime }$ end. Therefore, the beacon-labeled Strand A can be monitored in real time.
(2)$$2A\overset{k_{4}}{⟶}A$$

#### 2.3.2. Catalysis Reaction Module

The principle of this module is presented in Figure 3, and its mechanism can be described by the reactions listed in Equations (Equation 3) and (Equation 4).
(3)$$\left\{\begin{array}{c} A+B\overset{k_{5}}{⟶}C+W_{1}\\ C+D\overset{k_{6}}{⟶}2A+W_{2} \end{array}\right.$$
(4)$$A\overset{k_{7}}{⟶}2A$$

The process of the reaction can be described as follows: this module involves three main species, including Single-stranded A and Somplexes B and D, where A and D each carry a dye at their $5^{\prime }$ end, B carries a quencher at its $3^{\prime }$ end, and D carries a quencher at the end of $t_{1}^{*}$ (close to its $3^{\prime }$ end).

The toehold $t_{1}$ of A binds to the domain $t_{1}^{*}$ of B, and the branch migration moves gradually to domain $t_{3}$, which releases Single-stranded C together with double-stranded $W_{1}$. Then, toeholds $a_{1}$ and $a_{2}$ of C bind to the domains $a_{1}^{*}$ and $a_{2}^{*}$ of D, respectively, and the branch migration moves gradually to domain $t_{3}$, which releases double-stranded $W_{2}$ and two single-stranded A molecules. This implies that the concentration of A will be extended to twice its initial concentration.

### 2.4. Group Cyclic Shift

To extend the DNA-sequence-based initial key (Species A) so that it is sufficiently long, we introduce Algorithm 2 (to clearly describe these algorithms (Algorithms 2 and 3), we followed the way mentioned in [31,32,33]), hereafter referred to as groupCS, based on the group Cyclic Shift. For any sequence $S=s_{1}s_{2}\ldots s_{n-1}s_{n},$ let $O\left(S\right)=s_{2}\ldots s_{n-1}s_{n}s_{1},$ and $E\left(S\right)=\;s_{3}\ldots s_{n-1}s_{n}s_{1}s_{2}.$ For two sequences *S* and $S^{\prime }$, we denote by $S+S^{\prime }$ the resulting sequence obtained by connecting $S^{\prime }$ to *S* (at the end of *S*).
**Algorithm 2:** groupCS(D, $\ell_{0}$, *ℓ*), a procedure that extends a DNA sequence D of length $\ell_{0}$ to a new one of length at least *ℓ*.
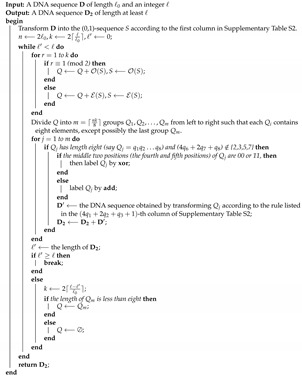


The algorithm first transforms the input DNA sequence D into a (0,1)-sequence *S* according to the first column in Appendix A (Line 2). Note that each base corresponds to a (0,1)-sequence of length 2; therefore, *S* has length $n=2\ell_{0}$. Then, a loop iteratively generates a DNA sequence $D_{2}$ with length at least *ℓ* (Lines 4–27).

In each iteration, a new (0,1)-sequence *Q* is constructed by *k* rounds of cyclic shift based on the current *S* (Lines 5–9), where the value of *k* is initially set as $2\lceil \frac{\ell }{\ell_{0}}\rceil$ and gradually decreases (Lines 3 and 23). To transform *S* into a DNA sequence, the algorithm divides *S* into $m=\lceil \frac{nk}{8}\rceil$ groups (say $Q_{1},Q_{2},\ldots ,Q_{m}$) from left to right such that each $Q_{i}$ contains eight elements, except possibly the last group $Q_{m}$, that is, when nk≠0 (mod eight), the last group contains less than eight elements (Line 10). Only a group of length eight (say $Q_{j}=q_{1}q_{2}\ldots q_{8}$) such that $\left(4q_{6}+2q_{7}+q_{8}\right)\notin \left\{2,3,5,7\right\}$ is transformed into the corresponding DNA sequence according to the rule listed in the $\left(4q_{1}+2q_{2}+q_{3}+1\right)$-th column of Appendix A (Lines 11–18). Next, if the length of $D_{2}$ is at least *ℓ*, then the algorithm breaks out of the loop and returns $D_{2}$; otherwise, *k* is reduced to $2\lceil \frac{\ell -\ell^{\prime }}{\ell_{0}}\rceil$, *Q* is updated by $Q_{m}$ or *∅*, and the algorithm implements the next iteration (Lines 19–27). We refer to the DNA sequence $D_{2}$ returned by groupCS as the *final key*. For examples of groupCS, refer to Appendix A.

### 2.5. The BioEN Algorithm

Based on the encryption framework and the above techniques, we developed a DNA-strand-displacement-based encryption algorithm (Algorithm 3), hereafter referred to as BioEN, which utilizes Huffman coding, DNA SDR, and cyclic shift. Note that the reverse process of BioEN is the corresponding decryption algorithm. This is illustrated by an example in Appendix A.
**Algorithm 3:** BioEN (a DNA-strand-displacement-based encryption algorithm).
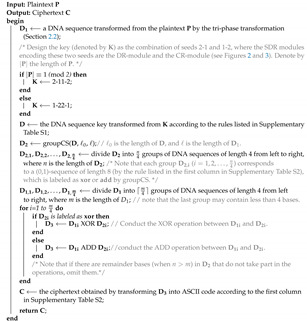


In light of the foregoing discussion, it is enough to explain how to transform $D_{3}$ into the final ASCII code, i.e., the ciphertext C (Line 18). First, transform $D_{3}$ into a (0,1)-sequence, denoted by *S*, according to the first column in Appendix A. Then, divide *S* into $k=\lceil \frac{t}{8}\rceil$ groups, from left to right, such that each group contains eight elements, except possibly the last group, where *t* is the length of *S*. Now, if the last group contains exactly eight elements, then add a new group consisting of eight zeros to *S* (at the end of *S*); if the last group contains less than eight elements, then add enough ones at the end of the last group so that the length of it is extended to eight, and add a new group of length eight consisting zeros or ones such that its corresponding decimal number is equal to the number of ones added to the last group. As a result, a (0,1)-sequence of length 8(k+1) is obtained, which can be divided into (k+1) groups, from left to right, such that each group contains eight elements. We refer to each of these groups as an *ASC-group*. Observe that the last ASC-group is used to identify how many ones are added, which serves for the decryption.

## 3. Validation of Feasibility

### 3.1. Experimental Setup

To show the feasibility of our approach in encryption, each experiment was set up with an experimental group and a control group. The concentration of the target DNA was expressed in the form of fluorescence intensity. The assembled DNA molecules were mixed according to the designed ratio, and the fluorescence intensity was monitored to obtain the final concentration of the target DNA strand.

All spectrofluorimetric measurements were performed using a real-time PCR system (QuantStudio 3 & 5 fluorescence quantitative PCR, Thermo Fisher Scientific, Waltham, MA, USA) equipped with a 96-well fluorescence plate reader. In the hold stage, the temperature was decreased by 1.6 ${}^{{}^{\circ}}$C to 4 ${}^{{}^{\circ}}$C/s and was then held for 10 s prior to the PCR stage. Then, the temperature was increased by 3 ${}^{{}^{\circ}}$C to 23 ${}^{{}^{\circ}}$C/s, and the fluorescence intensity was monitored every 10 s. The volume of each DNA sample was 20 μL.

### 3.2. Tools and Data

The sequences of all DNA strands in the experiment, listed in Appendix A, were designed by obtaining the original sequences using Nupack and then modifying the sequences by hand. The DNA oligonucleotides used were manufactured by Sangon Biotech (Shanghai, China). DNA oligonucleotides were purified by Sangon using high-performance liquid chromatography. Individual unlabeled DNA oligonucleotides were dissolved in 1 × TE buffer (nuclease free, pH 8.0, Sigma-Aldrich, St. Louis, MO, USA) and stored at −20 ${}^{{}^{\circ}}$C. Oligos labeled with dyes or quenchers were dissolved in deionized water (Milli-Q) and stored in deionized water at −20 ${}^{{}^{\circ}}$C. The DNA sample concentration was measured by *NanoPhotometer*${}^{®}$ N120 (Implen Inc., Westlake Village, CA, USA). All reagents were of analytical grade without further purification.

The DNA oligonucleotides were mixed in Tris-EDTA buffer (1×Tris-EDTA: 40 mM Tris base, 20 mM acetic acid, 2 mM EDTA adjusted to pH 8.0) with 12.5 mM MgCl_2_. All DNA complexes (listed in Appendix A) were mixed with an equal amount of corresponding single-stranded DNA to 10 μM. All samples were annealed in a polymerase chain reaction (PCR) thermal cycler. The temperature was set at 95 ${}^{{}^{\circ}}$C for 2 min initially and then decreased to 4 ${}^{{}^{\circ}}$C at a rate of −0.1 ${}^{{}^{\circ}}$C every 6 s. The hybridized molecules were stored at 4 ${}^{{}^{\circ}}$C for further use.

For simulation and dynamic analysis, we used Visual DSD [34]. The simulation duration was set to 600 s. The reactant concentration was at least 10 nM.

### 3.3. Experimentation Procedure

The initial key was obtained by biological experiments. Two DNA strand displacement modules were designed to obtain seeds 2-1 and 1-2. Before carrying out the biological experiments, simulation experiments were conducted as an auxiliary verification.

#### 3.3.1. Simulation Experiment of the DR-Module

In the Degradation Reaction (DR)-module, there were Single-stranded A and auxiliary Complexes B, D and G. The initial concentration of A was ${\left[A\right]}_{0}$= 20 nM, and the initial concentration of each of B, D, and G was $C_{m}$ = 10 nM. The (DSD) reaction rates $k_{1}=k_{2}=7\times 10^{-4}$/nM/s and $k_{3}=10^{-1}$/nM/s, where $k_{3}$ is the maximum reaction rate. The rate constants of the corresponding DNA reactions were determined according to the rate constants of the formal chemical reactions, which were equal to the rate constants of the corresponding DNA strand reaction multiplied by the initial concentration of the auxiliary complexes strands. $k_{1}$, $k_{4}$, and $C_{m}$ satisfy $k_{4}=k_{1}C_{m}.$ The simulation process was performed for 600 s, and the concentration of A was reduced from 20 nM to 10 nM (see Figure 4a).

#### 3.3.2. Biological Experiments of the DR-Module

To obtain the seed 2-1, we conducted two groups of biochemical reactions, named experiment group and control group, respectively, where the concentration of all species involved in the experiments (i.e., A, B, D, and G) was 10 μM, and the control group was just for reference. The experiment group included 4 μL of A, D, and G, respectively, and 6 μL of B, while the control group included 4 μL of A and 14 μL Tris-EDTA buffer (1× Tris-EDTA: 40 mM Tris base, 20 mM acetic acid, and 2 mM EDTA adjusted to pH 8.0). We put these two groups into the fluorescence quantitative PCR instrument and examined the fluorescence intensity change of A. Initially, they had the same concentration of A. When the reaction tended to be stable, the concentration of A in the experiment group was reduced by half, while the concentration of A in the control group was unchanged (see Figure 5a).

To show the key sensitivity of our approach (see Section 4.1.2), we conducted a contrast experiment, in which the experiment group included 5 μL of A, B, D, and G, respectively, while the control group included 5 μL of A and 15 μL Tris-EDTA buffer. The results are shown in Appendix A.

#### 3.3.3. Simulation Experiment of the CR-Module

In the Catalysis Reaction (CR)-module, there are single-stranded A and auxiliary complexes B and D. The initial concentration of A is ${\left[A\right]}_{0}$= 10 nM and the initial concentration of each of B and D is $C_{m}$= 10 nM. The (DSD) reaction rates $k_{5}=9\times 10^{-3}$/nM/s and $k_{6}=10^{-2}$/nM/s, where $k_{6}$ is the maximum reaction rate. The rate constants of the formal chemical reactions is equal to the rate constants of the corresponding DNA strand reaction multiplied by the initial concentration of the auxiliary complexes strands. $k_{5}$, $k_{7}$ and $C_{m}$ satisfy $k_{7}=k_{5}C_{m}.$ The simulation process was performed for 600 s, and the concentration of A was increased from 10 nM to 20 nM (see Figure 4b).

#### 3.3.4. Biological Experiments of CR-Module

To obtain the seed 1-2, we also conducted the two groups of experiments for the DR-module, in which the concentration of A, B, and D was 10 μM. The experiment group included 5 μL of A and D, respectively, and 6 μL of B, while the control group included 5 μL of A and 11 μL Tris-EDTA buffer. We put these two groups into the fluorescence quantitative PCR instrument and examined the fluorescence intensity change of A. Initially, they had the same concentration of A. When the reaction tended to be stable, the concentration of A in the experiment group was doubled, while the concentration of A in the control group was unchanged (see Figure 5b).

To show the key sensitivity of our approach (see Section 4.1.2), we conducted a contrast experiment, in which the experiment group included 7 μL of A, B, and D, respectively, while the control group included 7 μL of A and 14 μL Tris-EDTA buffer. The results are shown in Appendix A.

### 3.4. Experimental Results

The results are shown in Figure 4 and Figure 5, respectively. As expected, the simulation and biological experiment produced consistent results. This provides a guarantee for the performance of these two SDR modules, which can be used to encode the seeds 2-1 and 1-2, respectively.

## 4. Security Analysis

In this section, we analyzed the security of our encryption algorithm.

### 4.1. Key Sensitivity

An excellent encryption scheme should be sensitive to the key, meaning a minor change to the key will cause major changes to the results of encryption and decryption. Because our key is highly associated with biological experiments and the experiments are very sensitive to the environment, the desired key can be generated only when all experimental conditions are set correctly. Any mistake will lead to a different result, which implies that the key is sensitive. In addition, the key extension mechanism (groupCS) introduces considerable confusion to the final key. To illustrate this, we designed three types of experiments. The plaintext we used was “*anewencryptionapproachusingdnabiotechnologyandhuffmancoding*”, and the seed was 2-11-2.

#### 4.1.1. Change One Base

Referring to Appendix A, the seed 2–11–2 was transformed into the DNA sequence key D = GCCCGCAAGCCGGCCGGCAAGCCC. We wanted to investigate the difference of the encryption results (obtained by BioEN) when an arbitrary base in D is changed. In the experiment, we selected the fifth base G and changed it to T, i.e., the changed DNA sequence was $D^{\prime }$= GCCCTCAAGCCGGCCGGCAAGCCC. Based on D and $D^{\prime }$, the ciphertexts obtained by BioEN were completely different; see Figure 6a.

#### 4.1.2. Change Experiment Conditions

Note that when conducting the biological experiment, for the DR-module, the concentration ratio of Species A, B, D, and G was 2:3:2:2; and for the CR-module, the concentration ratio of A, B, and D was 5:6:5. To show the key is sensitive, we conducted a new experiment by setting the concentration ratio of A, B, D, and G to 1:1:1:1 for the DR-module and the concentration ratio of A, B, and D to 1:1:1 for the CR-module (see Appendix A for the results of the experiment). Consequently, the concentration changes of Species A for the DR-module and CR-module were 8:5 and 3:5, respectively, by which the seed we obtained was 8–53–5. Thus, the ciphertexts obtained by BioEN, based on the seeds 2–11–2 and 8–53–5, respectively, were very different; see Figure 6b.

#### 4.1.3. Change One Element in the Process of Extending the Key

To extend the seed 2-11-2, groupCS first transforms it into the DNA sequence D, which is further transformed to a (0,1)-sequence *S* by the rule listed in the first column of Appendix A. Then, based on *S*, a longer (0,1)-sequence *Q* is constructed according to the corresponding rules (Lines 6–9; groupCS). Note that here, we only considered the first iteration. We wanted to change an element of *Q* to test the effect on the final ciphertext. Thus, given the importance of each element’s position in *Q* (Lines 10–14; groupCS), we changed the eighth element of *Q* from zero to one, and all other elements remained unchanged. Figure 6c shows that even such a slight modification led to a significant change in the final ciphertexts.

### 4.2. Key Space Analysis

Note that the (0,1)-sequence *S* mentioned in Section 4.1.3 has length 48. Denote by R(S) the resulting (0,1)-sequence obtained from *S* by conducting the shift operation once, and let:$$R^{i}\left(S\right)=\underset{i\;\;Rs}{\underset{︸}{R(R(R(\ldots R}}\left(S\right))))$$
i.e., when i≡1(mod 2), $R^{i}\left(S\right)=O\left(R^{i-1}\left(S\right)\right)$; when i≡0(mod 2), $R^{i}\left(S\right)=E\left(R^{i-1}\left(S\right)\right)$, where $R^{0}\left(S\right)=S$ and *i* is a positive integer. Since the length of *S* is finite, there may exist some positive integer *r* such that $R^{r}\left(S\right)=S$ and $R^{r+i}\left(S\right)=R^{i}\left(S\right)$, where *i* is a nonnegative positive integer. We call the smallest *r* with this property the *rank* of *S*. Clearly, the final key is generated based on a (0,1)-sequence of length 48r, where *r* is the rank of *S*. We refer to the set of all distinct (0,1)-sequences of length 48r as the *key space* of the encryption algorithm BioEN. By a simple exhaustive analysis, we have the following proposition, which shows that the key space of BioEN is large enough to be secure. The detailed proof can be found in Appendix A.

**Theorem** **1.***The rank of S is 32, and the cardinality of the key space of BioEN is* 2^1536^.

### 4.3. Statistic Characteristic

We investigated the ASCII values of the characters appearing in the plaintext and ciphertext. Compared to the range of the ASCII values, we saw that the ASCII value distribution of the plaintext was 95–125, whereas that of the ciphertext was 0–255; see Figure 7. Such a large difference in ASCII values provides a strong guarantee for protection against statistical attacks.

## 5. Conclusions

We proposed a bio-experiment-based DNA encryption framework for data security (i.e., Algorithm 1). Based on the proposed framework, we introduced an encryption algorithm (i.e., BioEN) by designing a Huffman-coding-based method tri-phase transformation to deal with the unprocessed plaintext, two DNA SDR modules to generate the initial key, and a cyclic-shift-based mechanism (i.e., groupCS) to extend the key. The proposed algorithm highlights the importance of biochemical experiments. To validate the feasibility of the proposed algorithm, we conducted both a DSD simulation and a biochemical experiment. Compared to the existing DNA strand replacement encryption algorithms, the proposed algorithm is heavily dependent on the experiments and generates pseudo-random sequences by tracing the concentration change of the target DNA strand. Further analysis of the security showed that our algorithm is key sensitive, has a large key space, and can effectively resist statistical attacks. Compared with the works in [28,29], our encryption approach has the advantage of performing encryption through DNA strand displacement experiments rather than staying in the theory or simulation stage, which is expected to push forward the research of DNA-strand-displacement-based encryption. Though designed for text encryption, our encryption framework may be also applicable to image encryption or other areas of encryption, which would be worth exploring in future work. 

## Figures and Tables

**Figure 1 nanomaterials-12-00877-f001:**
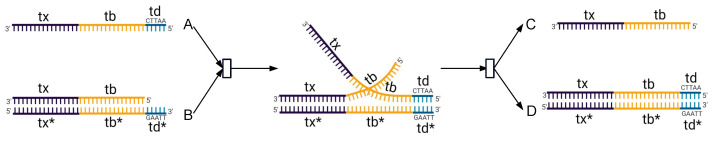
Principle of strand displacement reaction.

**Figure 2 nanomaterials-12-00877-f002:**
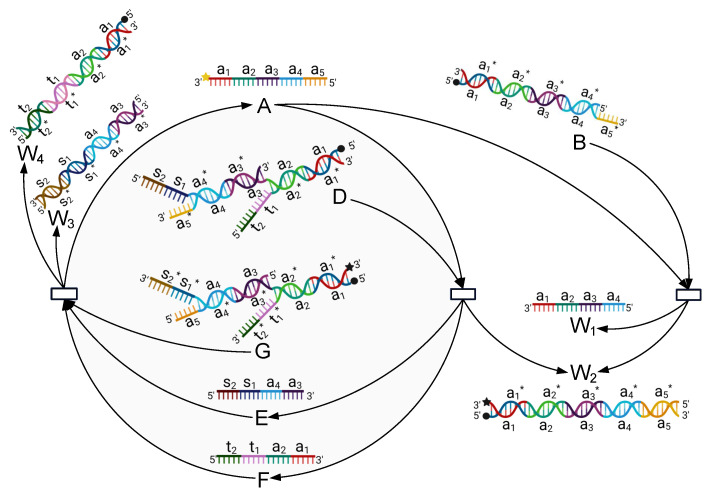
Schematic illustration of the degradation reaction module, by which the concentration of Species A is reduced to half of its original concentration. Thus, Species A can be used to encode the seed “2-1”.

**Figure 3 nanomaterials-12-00877-f003:**
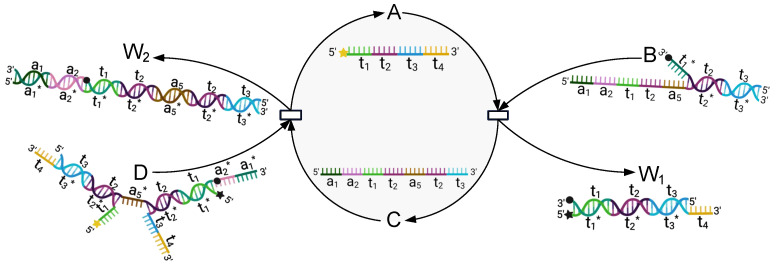
Schematic illustration of the catalytic reaction module, by which the concentration of Species A is extended to twice its original concentration so that Species A can be used to encode the seed “1–2”.

**Figure 4 nanomaterials-12-00877-f004:**
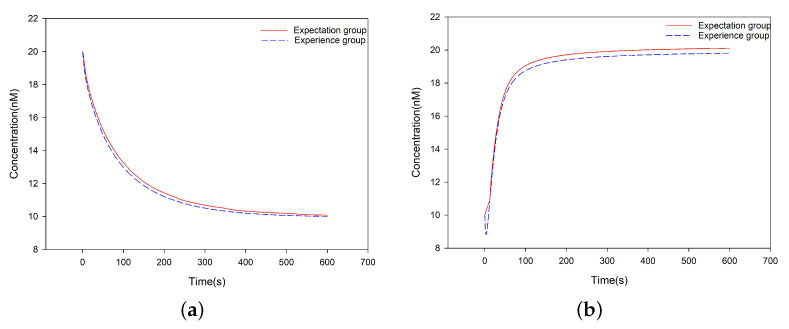
Simulation results. (**a**) The evolution of the concentration of Species A in the DR-module. The whole process takes 600 s, and the concentration of A is reduced from 20 nM to 10 nM. (**b**) The evolution of the concentration of Species A in the CR-module. The whole process takes 600 s, and the concentration of *A* is doubled.

**Figure 5 nanomaterials-12-00877-f005:**
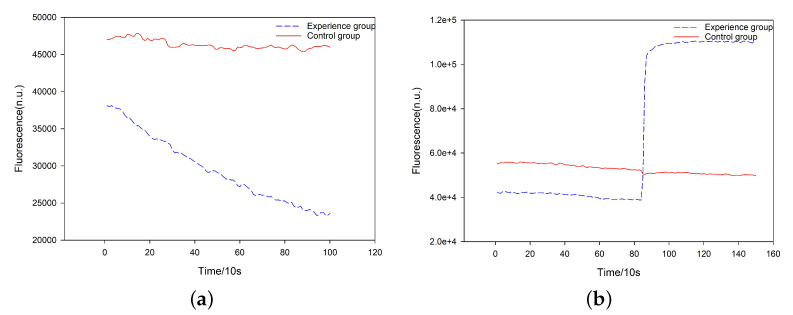
Biological experiment results. (**a**) The evolution of the concentration of Species A in the DR-module. (**b**) The evolution of the concentration of A in the CR-module. The concentration change of Species A in the experiment is the same as that in the simulation.

**Figure 6 nanomaterials-12-00877-f006:**
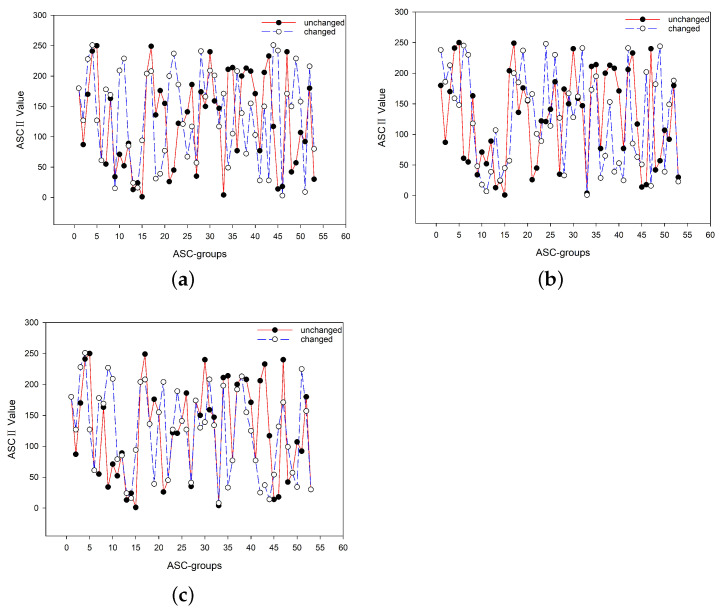
Key sensitivity tests, where the horizontal axis displays the specific ASC-groups, and the vertical axis presents the corresponding ASCII value of each ASC-group. The corresponding ASCII values exhibit great differences when factors related to the key are changed: (**a**) change one base; (**b**) change experimental conditions; (**c**) change one element in the process of extending the key.

**Figure 7 nanomaterials-12-00877-f007:**
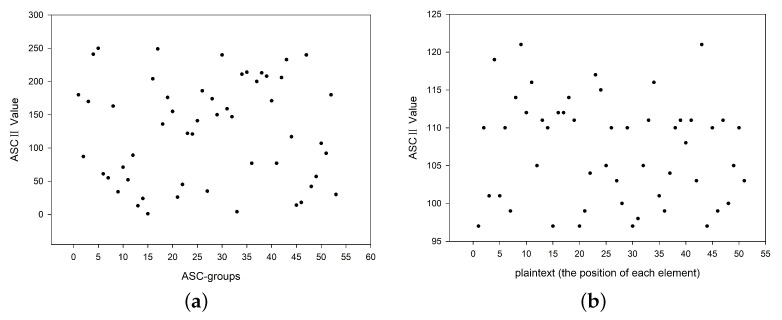
The ASCII value distribution of (**a**) the plaintext and (**b**) the ciphertext.

## Data Availability

The data presented in this study are available on request from the corresponding author.

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
