# Peer review of "An Operational DNA Strand Displacement Encryption Approach"

_nanomaterials, 2022, doi:10.3390/nano12050877_

Round 1

Reviewer 1 Report

The authors should revise what is trivial and what is not in their paper. Is unbalanced in this form, since "DNA" (molecules (?), encryption) is undefined while ADD, SUB, XOR and XNOR are explained (by the way, NXOR is better acronym instead of XNOR).

In the introduction part the authors need to justify the connection of their work with nanomaterials. Biochemical experiments are still a little far away.

The authors claims that they purchased biological material (DNA oligonucleotides) and they conducted a biological experiment of crossing ("2.2. Assembly procedure"). 

Despite of the fact that the manuscript presents an inspiring research, its materialization, unfortunately split in manuscript body, supplementary material and unpublishable material makes very difficult to assess the real contribution to the field.

The authors must clearly separate the (supporting) biological experiments, their input data and output results from the simulation and algorithms. Both the virtual space (the algorithms) and the real space (DNA strands) must meet together in a distinct section discussing (explicitly there) the (claimed new) unifying concepts. Otherwise, in my opinion, the current version of the manuscript looks like a mess in which nobody is able to separate the theoretical concepts, algorithms, from experiments (real experiments or virtual experiments) intended to support the use of the algorithms.

The style of writing the algorithms also can be improved. I realize that they are written at a certain level of unrefined syntax and it is very difficult to ask to be specified using a computational language (an example of probably too fully specified algorithms from the point of view of the authors is in 10.3390/math9192506) but - usually the algorithms are given for the specific purpose of illustrating the key part of a problem without having in mind language-specific limitations, but the authors must address a little improvement in their specification anyway - the purpose of  each algorithm as well as some comments along must be added. Probably is not a bad idea to take a look again and to add some - at least do the part of commenting the code. I wrote myself some genetic algorithms, but, again, if it is too complicated to give the algorithm in one shoot, please break the problem into subproblems (see the cited refs; see also Figure 2 and accompanying text in "Structure-activity relationships from natural evolution" online available paper).

Author Response

Dear Reviewer:

Thank you for your helpful comments, according to which we have improved our manuscript.  Our response and the corresponding modifications that have been made are as follows.

Question 1: The authors should revise what is trivial and what is not in their paper. Is unbalanced in this form, since "DNA" (molecules (?), encryption) is undefined while ADD, SUB, XOR and XNOR are explained (by the way, NXOR is better acronym instead of XNOR).

Response: Thank you for your suggestion. We have added an explanation on DNA in the new version (see first sentence of the Abstract; lines 10-11;).

For XNOR, it is an operation proposed by the article [18], where they name it XNOR. So we just follow this name.

Question 2: In the introduction part the authors need to justify the connection of their work with nanomaterials. Biochemical experiments are still a little far away.

Response: Thank for your comments. We have added a discussion on the connection of our work with nanomaterials and nanotechnologies (lines 40-43; lines 75-81).

Question 3: The authors claim that they purchased biological material (DNA oligonucleotides) and they conducted a biological experiment of crossing ("2.2. Assembly procedure"). 

Response: We appreciate your valuable suggestion. We have added an explanation on this issue (footnote 1, page 2; lines 294-297)

Question 4: Despite of the fact that the manuscript presents an inspiring research, its materialization, unfortunately split in manuscript body, supplementary material and unpublishable material makes very difficult to assess the real contribution to the field.

Response: Thank you for your suggestion. We have re-organized these materials in the manuscript and supplementary. The structure of the new version is illustrated in lines 136-146.

Question 5: The authors must clearly separate the (supporting) biological experiments, their input data and output results from the simulation and algorithms. Both the virtual space (the algorithms) and the real space (DNA strands) must meet together in a distinct section discussing (explicitly there) the (claimed new) unifying concepts. Otherwise, in my opinion, the current version of the manuscript looks like a mess in which nobody is able to separate the theoretical concepts, algorithms, from experiments (real experiments or virtual experiments) intended to support the use of the algorithms.

Response: Thanks a lot. The structure of the paper has been re-organized, especially sections 2,4,5 in the old version (sections 3, 4 in the new version).

Question 6: The style of writing the algorithms also can be improved. I realize that they are written at a certain level of unrefined syntax and it is very difficult to ask to be specified using a computational language (an example of probably too fully specified algorithms from the point of view of the authors is in 10.3390/math9192506) but - usually the algorithms are given for the specific purpose of illustrating the key part of a problem without having in mind language-specific limitations, but the authors must address a little improvement in their specification anyway - the purpose of  each algorithm as well as some comments along must be added. Probably is not a bad idea to take a look again and to add some - at least do the part of commenting the code. I wrote myself some genetic algorithms, but, again, if it is too complicated to give the algorithm in one shoot, please break the problem into subproblems (see the cited refs; see also Figure 2 and accompanying text in "Structure-activity relationships from natural evolution" online available paper).

Response: We gratefully appreciate for your valuable suggestion. We have improved the presentation style of the long algorithm with reference to your suggestions. (Algorithm 3, page 8)

Reviewer 2 Report

The article aims to present a Bio-Experiment-Based Encryption Approach. However, the uniqueness of the work is not evident from the title. Therefore, it is suggested to modify the title so that the proposed work may be differentiated from existing Bio-Experiment-Based Encryption methods. 

The article is well written and easy to follow. However, the following issues can be addressed before publication: 

(1) Abstract: 

Describe the target problem and its importance.

Limitations of existing Bio-Experiment-Based Encryption Approaches.

While the main features of the proposed method are clearly explained, it is important to mention that how authors have validated the solution by describing the used benchmarks and case studies.

The outcomes, significance and impact of the proposed solution.

(2) Introduction:

The research problem is nicely presented by explaining what is really required in the selected research area. A briefly overview of the existing practices has been presented.  However, limitations of existing practices  are not identified In other words, the research gape is not identified.

The proposed solution is described in a logical way. However, it is not clear that how the proposed method is going to address the identified research gap.  How it will solve the limitations ?   Moreover, the novelty of proposed solution should be highlighted.

How authors have validated the solution (description of case studies and/or benchmarks. Why these benchmarks/case studies are important and interesting. What is the motivation behind the selection of these benchmarks)

Achieved results as well as significance and impact of the solution should be discussed

(3) Proposed Approach:

--It is a very well written section and the proposed method has been described with some useful figures   such that each figure has been explained properly in the text. 

(4) Validation:   

This section is really the place where authors should describe that how they have implemented the solution. It may contains the following subsection (as per the requirements of the paper): 

--Experimental setup

--Description of tools and particular settings  used in the experiment

--Description of Benchmarks and case study.

--Experimentation Procedure: 

--Experimental Results: 

(5) Performance Comparison and Discussion  

Compare the achieved results (obtained in the previous section) with state-of-art solutions, using tables, graphs and any other methods for comparison. Good data visualization techniques may enhance the value of the article.

Use various performance parameters for comparisons. Each performance parameter must be defined explicitly before its use. Authors should also provide the justification and motivation behind the selection of each performance parameter.

It is VERY IMPORTANT to highlight the strengths of the article as compared to existing methods. Discuss the reasons that why authors were able to obtain good/better results in terms of certain performance parameters. 

 Similarly, it is equally important to mention the shortcomings of the proposed solution with respect to current methods. Discuss the reasons for these limitations and shortcomings.

Author Response

Dear Reviewer,

Thank you for your comments and advice, according to which we have improved the manuscript. A point-by-point response is given below along with the modifications that have been made.

(1) Title:

Question 1. The article aims to present a Bio-Experiment-Based Encryption Approach. However, the uniqueness of the work is not evident from the title. Therefore, it is suggested to modify the title so that the proposed work may be differentiated from existing Bio-Experiment-Based Encryption methods. 

Response: Thank you for pointing out this problem. To be able to highlight the uniqueness of the work, the title has been changed as: An Operational DNA Strand Displacement Encryption Approach. (With the new title, the proposed work can be distinguished from existing encryption methods since the proposed approach is operational via biological experiments based on DNA strand displacement technology.)

(2) Abstract:

Question 2.1. Describe the target problem and its importance.

Response: Thank you for the above suggestion. In the revised manuscript, we have modified the abstract to introduce the target issue and its importance. (lines 10-13)

Question 2.2. Limitations of existing Bio-Experiment-Based Encryption Approaches.

Response: Thanks a lot. We have also added a description of the limitations of existing DNA encryption approaches. To the best of our knowledge, this is the first work on biologically operational DNA encryption based on DNA strand displacement. (lines 13-16)

Question 2.3. While the main features of the proposed method are clearly explained, it is important to mention that how authors have validated the solution by describing the used benchmarks and case studies.

Response: Thank you. We have re-written this part to make validation part clearer. (lines 25-27).

Question 2.4. The outcomes, significance and impact of the proposed solution.

Response: We have re-written this part according to your suggestion. (lines 27-29)

(3) Introduction:

Question 3.1. The research problem is nicely presented by explaining what is really required in the selected research area. A briefly overview of the existing practices has been presented.  However, limitations of existing practices are not identified In other words, the research gape is not identified.

Response: We greatly appreciate your valuable suggestion. We have added the discussions on limitations of existing works in the Introduction (lines 105-111).

Question 3.2. The proposed solution is described in a logical way. However, it is not clear that how the proposed method is going to address the identified research gap.  How it will solve the limitations ?   Moreover, the novelty of proposed solution should be highlighted.

Response: Yes, we have added a description on the contribution and innovation of the paper.  (lines 109-125)

Question 3.3 How authors have validated the solution (description of case studies and/or benchmarks. Why these benchmarks/case studies are important and interesting. What is the motivation behind the selection of these benchmarks)

Response: We have re-written this part according to your good suggestion. (lines 126-133).

Question 3.4 Achieved results as well as significance and impact of the solution should be discussed

Response: Thank you. We have added the discussion on the significance and impact of our approach.  (lines 133-135).

(4) Proposed Approach:

Question 4. It is a very well written section and the proposed method has been described with some useful figures   such that each figure has been explained properly in the text. 

Response: Thank you.

(5) Validation: 

Question 5. This section is really the place where authors should describe that how they have implemented the solution.

Response: Thanks. We have re-organized the materials concerning the experiments, and have re-written this chapter to present clearly the implementation of the approach. It consists of several subsections including Experimental setup, Tools and data, Experimentation Procedure and Experimental Results as you suggested.

(6) Performance Comparison and Discussion

Question 6.1 Compare the achieved results (obtained in the previous section) with state-of-art solutions, using tables, graphs and any other methods for comparison. Good data visualization techniques may enhance the value of the article.

Question 6.2 Use various performance parameters for comparisons. Each performance parameter must be defined explicitly before its use. Authors should also provide the justification and motivation behind the selection of each performance parameter.

Question 6.3 It is VERY IMPORTANT to highlight the strengths of the article as compared to existing methods. Discuss the reasons that why authors were able to obtain good/better results in terms of certain performance parameters. 

 Question 6.4 Similarly, it is equally important to mention the shortcomings of the proposed solution with respect to current methods. Discuss the reasons for these limitations and shortcomings.

Response 6.1-6.4: For this issue, we would like to make an explain to the reviewer. Since (to the best of our knowledge) this work is the first to achieve encryption through DNA strand displacement experiments. The previous studies on DNA strand displacement-based encryption are mainly staying at the stage of theory or simulation, which is difficult to realize through biological experiments since real-time monitoring of the concentration of target DNA strands are required in their framework. Therefore, the main purpose of the experiment in this work is to show the feasibility of this approach, which is an important step forward. And thus no experiments on the performance comparison of our approach and the existing approach can be done. The strength of our approach is that bio-experiments are achievable under this framework. We have added a discussion on this. (lines 436-439).

Round 2

Reviewer 1 Report

In my opinion the manuscript is ready for publication.

Reviewer 2 Report

The comments have been addressed.

The article can be published in its current form.